# Atlas of Nervous System Vascular Malformations: A Systematic Review

**DOI:** 10.3390/life12081199

**Published:** 2022-08-07

**Authors:** Carlos Castillo-Rangel, Gerardo Marín, Karla Aketzalli Hernandez-Contreras, Cristofer Zarate-Calderon, Micheel Merari Vichi-Ramirez, Wilmar Cortez-Saldias, Marco Antonio Rodriguez-Florido, Ámbar Elizabeth Riley-Moguel, Omar Pichardo, Osvaldo Torres-Pineda, Helena G. Vega-Quesada, Ramiro Lopez-Elizalde, Jaime Ordoñez-Granja, Hugo Helec Alvarado-Martinez, Luis Andrés Vega-Quesada, Gonzalo Emiliano Aranda-Abreu

**Affiliations:** 1Department of Neurosurgery, “Hospital Regional 1º de Octubre”, Institute of Social Security and Services for State Workers (ISSSTE), Mexico City 07300, Mexico; 2Biophysics Department, Brain Research Institute, Xalapa 91192, Mexico; 3Molecular Biology and Cell Culture, Brain Research Institute, Xalapa 91192, Mexico; 4National Center of Medicine, “Siglo XXI: Dr. Bernardo Sepúlveda Gutiérrez”, Mexican Social Security Institute (IMSS), Mexico City 07300, Mexico; 5Department of Internal Medicine, General Hospital of Zone No. 71 “Lic. Benito Coquet Lagunes”, Veracruz 91700, Mexico; 6General Hospital of Zone No. 17, Mexican Institute of Social Security (IMSS), Ciudad de México 06600, Mexico; 7Department of Cardiology, Cardiology Hospital of Zone No. 34, Mexican Institute of Social Security (IMSS), Ciudad de México 06600, Mexico

**Keywords:** brain vascular anomalies, cerebral vascular malformations, neuroimaging, approach, atlas

## Abstract

Vascular malformations are frequent in the head and neck region, affecting the nervous system. The wide range of therapeutic approaches demand the correct anatomical, morphological, and functional characterization of these lesions supported by imaging. Using a systematic search protocol in PubMed, Google Scholar, Ebsco, Redalyc, and SciELO, the authors extracted clinical studies, review articles, book chapters, and case reports that provided information about vascular cerebral malformations, in accordance with Preferred Reporting Items for Systematic Reviews and Meta-Analyses (PRISMA) guidelines. A total of 385,614 articles were grouped; using the inclusion and exclusion criteria, three of the authors independently selected 51 articles about five vascular cerebral malformations: venous malformation, brain capillary telangiectasia, brain cavernous angiomas, arteriovenous malformation, and leptomeningeal angiomatosis as part of Sturge–Weber syndrome. We described the next topics—“definition”, “etiology”, “pathophysiology”, and “treatment”—with a focus on the relationship with the imaging approach. We concluded that the correct anatomical, morphological, and functional characterization of cerebral vascular malformations by means of various imaging studies is highly relevant in determining the therapeutic approach, and that new lines of therapeutic approaches continue to depend on the imaging evaluation of these lesions.

## 1. Introduction

Vascular anomalies comprise a heterogeneous spectrum of structural disorders of blood and lymphatic vessels that are present in 4.5% of the world’s population at birth and during childhood [1]. These anomalies are classified into two groups: tumors and vascular malformations. Tumors are characterized by hyperplasia that generally conditions an accelerated growth of said structure, unlike vascular malformations, whose cells are in quiescence, implying a lack of structure growth and to a lesser extent a gradual increase in size. Specifically, vascular malformations never involute; they are believed to be derived from alterations in the angiogenesis process during the embryonic stage, as well as from alterations in the development of the vessels during the extrauterine life [2,3,4]. They are generally derived from germline or somatic genetic alterations [5]. The main classification system for vascular anomalies. for international standardization purposes that establish clinical and histopathological criteria, is the one proposed by the International Society for the Study of Vascular Anomalies (ISSVA), which was updated in 2018 [6].

Vascular malformations are divided into four groups, according to the ISSVA. The first group includes simple vascular malformations, which involve only capillaries (9%), veins (70%), or lymphatic vessels (15%). This group includes, exceptionally, arteriovenous malformations (AVM) (6%). The second group corresponds to combined vascular malformations that involve two or more of these structures [2,3]. The third group includes malformations that are associated with other anomalies, such as CLAPO (capillary vascular malformation of the lower lip, lymphatic malformation of the head and neck, asymmetry, and partial or generalized overgrowth) and CLOVES (congenital lipomatous overgrowth, vascular malformations, epidermal nevi, and skeletal anomalies). The last group includes malformations that affect large caliber vessels, such as arteries, lymphatic vessels, or veins [6,7,8]. 

Functionally, vascular malformations are divided into low flow or high flow. Low flow lesions are usually capillary malformations, venous malformations (VMs), and lymphatic malformations (LMs). High flow lesions involve the presence of arterial vessels, and include arteriovenous malformations (AVMs) and fistulas [3,9,10].

Between 40% and 60% of vascular anomalies occur in the head and neck region, mostly affecting the nervous system [2,11]. The way to approach these anomalies depends on the specific characteristics that they exhibit [8]. Therefore, it is important to know the main characteristics of simple group vascular malformations, according to ISSVA criteria, which are present in the nervous system, to achieve correct identifications, characterizations, and therapeutic approaches (Figure 1). This information is obtained by clinical anatomical imaging (angiography and magnetic resonance imaging) and histopathological analysis. 

## 2. Materials and Methods

### 2.1. Search Strategies

A systematic electronic search was conducted in accordance with the Preferred Reporting Items for Systematic Reviews and Meta-Analyses (PRISMA) [12] guidelines in the databases of PubMed, Google Scholar, Ebsco, Redalyc, and Scielo, with a focus on clinical studies, review articles, book chapters, and case reports. that search provided information about simple vascular malformations in the nervous system, in accordance with the 2018 guidelines of the International Society for the Study of Vascular Anomalies (ISSVA) [6]. All databases were last consulted in June 2022. Database-specific filters were used, as necessary, to complete the searches in all the specified databases. the search strategies and keywords are illustrated in Figure 2.

### 2.2. Study Selection

Using the inclusion and exclusion criteria, three of the authors independently screened the titles and abstracts of the retrieved studies to determine the ones that required further assessment. After duplicates were removed, the authors further assessed the potential studies that were identified via the search strategy. When relevance was determined, the full texts of the articles were retrieved and assessed for possible inclusion, based on the relationship of the contents to the “definition”, “etiology”, “pathophysiology”, and “treatment” of each vascular malformation in the nervous system, and on the relationship of the articles to the imaging approach. The study procedure is depicted in a PRISM flow diagram (Figure 2).

## 3. Results

The authors selected 56 articles for the analysis (See Appendix A), with a focus on a total of five vascular malformations that belong to a simple group of anatomic classification in accordance with the 2018 ISSVA guidelines: venous malformations (cerebral venous malformations and brain cavernous angiomas), arteriovenous malformations, and capillary malformations (brain capillary telangiectasia and leptomeningeal angiomatosis as part of the Sturge–Weber syndrome). Forty-three clinical studies and review articles, two book chapters, and 11 case reports were included. Each subtopic was developed via a synthesized description based on data obtained from clinical studies, review articles, book chapters, and case reports. Illustrative images of anatomic, imaging, and histopathologic features were also selected. In addition, original images and schematic illustrations were added (Figure 3).

### 3.1. Venous Malformations

#### 3.1.1. Cerebral Venous Malformations

Cerebral venous malformations (CVMs) are sets of veins and venules that are arranged radially. This part of the structure is called the *nidus*. It is composed of a single endothelial layer surrounded by a discontinuous and irregular layer of myocytes and partially isolated from the surrounding veins. It is considered to be a low-flow malformation, frequently manifested during middle or late childhood [14,15,16,17]. AVMs are associated with congenital alterations resulting in the PIK3CA and TEK/TIE2 phenotypes. The latter is an endothelial cell-specific tyrosine kinase receptor that functions through the phosphatidylinositol 3-kinase (PI3K)/protein kinase B (AKT)/mammalian target of rapamycin (mTOR) (PI3K/AKT/mTOR) signaling pathways. It is believed to cause 50% of sporadic venous malformations and has been linked to multifocal venous malformations to a lesser extent [10,18]. Ninety-four percent of the cases are de novo mutations; 5% are glomuvenous malformations of dominant and non-hereditary inheritance; and 1% are cutaneomucosal malformations of dominant inheritance [16]. 

VMs lack rigidity. Therefore, they are compressible, and their volume depends on variations in systemic and local blood pressure. In some cases, they can be identified by symptomatology when performing Valsalva maneuvers. This is particularly the case for large volume malformations (>10 mL) or those that show phleboliths or belong to multifocal diseases, or those that are exhibited by patients with a history of Klippel–Trenaunay syndrome, suggesting a greater probability of developing intravascular coagulopathy [14,15,19]. Intravascular coagulopathy can be identified by D-dimer elevation, decreased fibrinogen, and low platelet count (100.000 to 150.000/µL), without decreased prothrombin time (PT) and activated partial thromboplastin time (PTT). In turn, this increases the risk of developing systemic coagulopathy in the event of sudden changes in intraluminal blood pressure [14,15,16]. Variants, such as developmental venous malformation, are usually asymptomatic; however, due to blood vessel fragility and the large volume that these malformations can reach, hemorrhage may occur. 

It should be noted that manifestations may vary, because they depend on a lesion’s location; however, headache and seizures are the most frequent symptoms. Specifically, when the location is infratentorial, such as in the middle cerebellar peduncle, neurosensory hearing loss has been reported, followed by facial paresis, due to the anatomical relationship of the lesion’s location with respect to cranial nerves VII to VIII. However, this is an extremely rare manifestation [20]. It should be underscored that this malformation’s main location is where the superficial and deep supratentorial blood drains converge, usually adjacent to the cortical or ependymal surface; they occur in the cerebellum to a lesser extent [17]. The main differential diagnosis is infantile hemangioma, which, unlike VMs, tends to involute [8].

The first-line imaging modality for diagnostic purposes is via Doppler ultrasound, as long as deep-seated and extensive lesions are not involved. The obtained diagnostic orientation data are low-flow or stasis in a multiple tubular structure, and are associated with a heterogeneous internal coloration. As a second-line modality, magnetic resonance imaging (MRI) is recommended, using conventional spin-echo (SE) sequences with T1-weighted and T2-weighted and fat-suppression images. Vascular malformations, such as hyperintense structures with serpiginous tubules, are observed in T2, as is heterogeneous intensity in the cases of hemorrhage or thrombosis. The specificity of the diagnosis increases with the identification of phleboliths, which are observed as hypointense areas that may be better visualized via computerized tomography (CT) [8,15]. In fact, the CT method allows for the characterization of the nidus size, the size and number of afferent and efferent circulatory vessels, or determining whether there is a connection with a deep venous system [8]. MRI protocols with contrast medium and vascular techniques (angiography and venography) allow for the evaluation and characterization of the malformation, whereby the angiogram shows the pathognomonic sign referred to as “caput medusa” (Figure 4) [8,15,17].

In general, sclerotherapy is the first-line treatment for symptomatic VMs. It blocks venous drainage by inducing structural modification of the blood vessel via scarring. Therefore, the risk of thrombosis must be assessed and prevented. Given that this procedure decreases the volume of the malformation, it is recommended as a preparation measure prior to surgical treatment in the case of large lesions, or as a single treatment in the case of smaller and easy-access lesions, because the procedure implies a lower incidence of external scars and aesthetic repercussions in comparison with surgical resection [8,14].

The most commonly used sclerosing agent is sodium tetradecyl sulfate (STS), followed by doxycycline, bleomycin, ethanolamine oleate, polidocanol, and absolute ethanol. Absolute ethanol is increasingly less used as a sclerosing agent. When multiple sclerotherapy interventions may be required, the determining factors for the choice of a sclerosing agent are the malformation’s anatomical characteristics and the medical practitioner’s experience [8]. Surgical resection is considered for patients with small AVMs, which can be completely removed, or larger AVMs with well-defined margins; however, the choice of a surgical approach should consider the high incidence of morbidity and recurrence. Preoperative embolization with n-BCA glue decreases both the risk of recurrence and the size of the malformation; thus, it is suggested mainly in cases of large VMs with poorly defined margins [8,14].

#### 3.1.2. Brain Cavernous Angiomas

Brain cavernous angiomas (CAs), or cavernous malformations, consist of a set of capillaries that lack the smooth muscle layer. They are composed of only two thin and permeable layers: an adventitious layer and an endothelial layer. These capillaries are usually large, full of blood, and isolated from each other by the extracellular matrix, resulting in their macroscopic “blackberry” shape. The size of these malformations can vary from 2 mm to several centimeters in diameter [21,22,23]. It is believed that in 70% of the cases, brain parenchyma cells are included among the channels of the Cas’ structure [24]. Depending on the etiology, they may appear either as isolated (sporadic) or multifocal lesions [25].

CAs are believed to be the result of mutations that are linked to an autosomal dominant (familial) or de novo pattern, specifically due to deletions, changes in the reading frame, and changes and in the splice site of the CCM1 (KRIT1), CCM2 (malcavernin, MGC4607), or CCM3 (PDCD10) genes [25,26] The latter is mainly associated with multifocal appearance [10]. Proteins encoded by these genes are components of a heterotrimeric intracellular adapter protein complex called the CCM complex, which contributes to the regulation of cell-to-cell interaction functions, adhesion, and the proliferation of endothelial cells through the activity regulation of the MEKK3-KLF2/4 and RhoA/Rho kinase (ROCK) pathways [27,28].

The etiology derived from de novo mutations has been related to differences in enteric microbiome, as it has been observed that patients with CAs have a higher proportion of Gram-negative bacteria, among which *Odoribacter splanchnicus* stands out, and a lower ratio of Gram-positive bacteria, such as *Faecalibacterium prausnitzii* and *Bifidobacterium adolescencia*. The lipopolysaccharides of Gram-negative bacteria are able to activate the CD14 and TLR4 receptors that are involved in the activation of the MEKK3 pathway, interfering with the regulation of the negative activity of the MEKK3-KLF2/4 pathway that is exerted by the CMM complex. This allows for the consideration of the influence of the intestinal-brain axis on the development and progression of CAs [28,29,30].

Although most CAs are small and asymptomatic [31], commonly exhibited manifestations are seizures [32,33], intracranial hemorrhage, focal neurological deficits (FNDs) [24], iron and hemosiderin deposits, gliosis, B- and T-cell infiltration, plasma cells and oligoclonal immunoglobulins, and complement activation [34]. These deficits cause neuroinflammation and calcification and, in the case of large lesions, ossification [25]. However, depending on the location of the lesions, specific ailments may be found, as detailed in Table 1**.**

Given that ACs lack feeding arteries and drainage veins, they are considered to be angiographically occult vascular malformations. Therefore, the first-line diagnostic tool is the MRI study using the susceptibility-weighted imaging (SWI) protocol, due to its high sensitivity in detecting deoxygenated hemoglobin and hemosiderin deposits that are not usually accurately visible in the T1 and T2 protocols. the second line, in terms of sensitivity for the detection of Cas, corresponds to the T2-weighted echo gradient protocol (GRE T2). However, the hemosiderin ring “blooming” artifact that is associated with CAs can maximize the estimation of lesion size in GRE T2. Therefore, T1 and (mainly) T2 sequences are more useful in estimating the size of CAs once the diagnosis has been established. On multiple occasions, the obtained images turn out to be similar to arteriovenous malformations (AVM). Therefore, in order to establish a differential diagnosis, digital subtraction angiography (DSA) via catheter may be used [17,40].

The first-line treatment for CAs consists of surgical resection of the lesion. The hemosiderin ring, the gliosis area and neuroinflammation surrounding the CAs are the resection area’s boundaries. Stereotactic radiosurgery has also been proposed as an alternative for patients with surgically inaccessible lesions [33]. Although a pharmacological treatment for CAs has not been standardized, preclinical studies have reported good results with atorvastatin that inhibits ROCK [22] and propranolol, whose therapeutic target is VEGF. It acts as an angiogenesis regulator, and therefore decreases the size of CAs (Figure 5 and Figure 6) [41].

### 3.2. Arteriovenous Malformation (AV Angioma, Cirsoid Angioma)

Arteriovenous malformations (AVMs) or arteriovenous hemangiomas consist of the direct connection between an artery and a vein, without intermediate capillaries. Therefore, they are classified in the high-flow group. The vessels that make up this malformation are dilated and dysplastic. They are jointly referred to as the nidus, which is connected with feeding arteries and drainage veins without the presence of brain parenchyma cells interspersed between the nidus vessels [5,42,43,44]. These connections allow AVMs to exert circulatory arrest on the surrounding brain structures [45]. The main irrigation arteries belong to the internal carotid or vertebrobasilar systems and the external carotid arteries, such as the middle or posterior meningeal artery [44]. Furthermore, the veins communicate with the drainage system of the adjacent brain structures [45]. AVMs are often pyramid-shaped, with the base facing the meninges and the vertex facing the ventricles [17,46]. Other less common morphology variants of AVMs are “wedge”, cylindrical, or globoid shapes restricted to white matter [44].

AVMs are derived from somatic mutations of the KRAS, NRAS, HRAS, BRAF, and MAP2K1 genes [10]. The latter three genes have been associated with the sporadic occurrence of AVMs representing 95% of cases [46,47] while the remaining 5% of the cases are attributed to dominant autosomal familial syndromes. In some of them, the NRAS gene is involved, as well as non-syndromic conditions, such as somatic mutations in the KRAS proto-oncogene [46]. Risk factors have been identified that coexist with AVMs, such as previous intracranial hemorrhage from trauma, moyamoya disease, radiation therapy, and epilepsy [48].

Regardless of the etiology of AVMs, their development coincides with errors in vascular morphogenesis in the primitive vascular plexus, involving angiogenic growth factors and their receptors, retained vascular connections, vascular changes due to inflammatory responses, and increased vascular endothelial growth factor preventing the development of capillary beds [17,44,49]. These alterations lead to arterial dilation, increased intraluminal pressure, venous infarction, intracranial hemorrhages, and increased venous pressure, leading to manifestations that depend on the affected brain region, including headache, tinnitus, dizziness, vertigo, paraesthesia, irritability, seizures and/or epilepsy events, visual problems, gait disturbance, instability, and neurological deficits, as well as intellectual disabilities, language disorders, ataxia, and dyslalia [49,50,51].

The initial diagnostic approach to this malformation can be supported by ultrasound study to observe anechoic tubular structures that are channeled toward a central nidus, and by the Doppler protocol to observe the flow in these structures, which are often turbulent, especially in the nidus, accompanied by arterialization of drainage veins and high-speed and low-resistance waves. From these studies, it is important to be able to quantify the feeding and drainage vessels, as well as the size of the VAMs and, particularly, the size of the nidus. The MRI via T2 protocol facilitates observation of dilated meandering vessels with flow gaps in connection to edema and without mass effect. The lesion size can be estimated more accurately via four-dimensional [4D] MR angiography with dynamic contrast. Additionally, this study allows for a more accurate observation of the lesion’s anatomical characteristics and the effect it exerts on surrounding structures. In particular, the DSA study shows the central conglomerate that corresponds to the nidus, surrounded by veins that exhibit proximal opacification. This allows for a differential diagnosis with CAs (Figure 7 and Figure 8) [8].

The treatment consists of microsurgical resection, robotic stereotactic radiosurgery (RSR), endovascular embolization, and combined multimodal management. Microsurgery is recommended mainly in AVMs with a superficial location, which lack deep venous drainage and have a size of less than 6 cm. In the case of superficial AVMs with a size of less than 3 cm, SRS is recommended It offers an obliteration incidence of 59% to 68%, although the risk of radiogenic edema development and the low risk of radionecrosis must be considered. It should be noted that the results have been more favorable when a sclerotherapy session is added prior to the intervention through SRS [52]. Endovascular embolization is considered to be suitable for AVMs with a size greater than 3 cm, when the intent is to reduce the size and to be able subsequently to practice microsurgery or radiosurgery. Combined multimodal embolization measures promotes angiogenesis that is secondary to hypoxia. This has been associated with greater recurrence of AVMs following an embolization-only approach [49,52,53,54,55].

### 3.3. Capillary Malformations

#### 3.3.1. Cerebral Capillary Telangiectasia

Cerebral capillary telangiectasia (CCT) consists of dilated, low-flow vascular channels of with varying diameters between 3 mm and 20 mm. CCT is composed of thin walls of endothelial cells, lacking a layer of smooth muscle cells. CCT is arranged in confluent groups and interspersed in the cerebral parenchyma [56,57,58]. This malformation is characterized by the lack of surrounding gliosis, calcification, macrophages loaded with hemosiderin, or mass effect. The relevant data allow for differential diagnosis through brain cavernous angioma (CA) [17].

The etiology of CCT is associated with a mutation of the GNAQ gene, the RASA1 gene, or the MAP2K1 gene.Mutations of the latter two genes are associated with the coexistence of CCT and arteriovenous malformations (AVMs) [18]. The symptomatology of CCT is heterogeneous. In some cases, it appears as a clinically silent lesion, while symptomatic cases are classified into three large groups. The first group consists of cerebral hemorrhages that are derived from capillary fragility in conjunction with venous hypertension events. The second group consists of a focal neurological deficit that is secondary to local stasis and thrombosis. The third group consists of all of those variable manifestations that are secondary to capillary and venous ectasia. The most frequent manifestations consist of cranial nerves and, in some cases, epileptic syndromes. Another factor that can influence the manifestations is the lesion’s location. The most commonly affected regions are the protuberance, the insular cortex, the midbrain, the uncus, the cerebellum, and the spinal cord [58].

In asymptomatic cases, lesions can show up as incidental findings during conventional MRI sequences. They are observed as isolated isointense lesions. The protocols with the use of gadolinium allow for them to be differentiated from other vascular anomalies, mainly CA which is very similar in imaging studies, showing an elevation of the edges in the form of a brush, a “dotted” appearance, or multiple dotted foci that arecharacteristic of CCT. The enhancement is due to the contrast medium present in ectasic vessels. However, the susceptibility-weighted imaging (SWI) study is of greater sensitivity for diagnosis, as it provides a submillimeter definition (Figure 9 and Figure 10) [58].

Asymptomatic patients may be treated conservatively, due to the extremely low risk of the progression or the bleeding of these lesions. Therefore, a precise radiographic diagnosis is required to determine the therapeutic approach and to avoid confusing them with other more serious alterations. Because of this, it is feasible to carry out a genetic study to rule out hereditary hemorrhagic telangiectasia when the results of the imaging studies and the clinical data are not conclusive [58,59].

In particular, in cases where the lesion is accessible and jeopardizes functional integrity, surgical removal has been recommended [58,59]. Although this procedure has been reported to improve neural functions or to reduce seizures in some clinical cases [17,60], endovascular treatments are listed as equally appropriate approaches [8,14]. The use of aminocaproic acid or tranexamic acid appears to inhibit fibrinolysis in the case of hemorrhagic complications [61]. Although CCT is believed to be a clinically benign entity, its timely diagnosis and treatment are essential in avoiding unnecessary invasive procedures, as well as in the development of neurodegenerative processes [17,59].

#### 3.3.2. Capillary-Venous Angioma (Sturge–Weber)

Sturge–Weber syndrome (SWS) or encephalotrigeminal angiomatosis, is a non-hereditary congenital syndrome, classified as neurocutaneous phacomatosis. It is characterized by capillary–venous malformation of the brain, glaucoma, seizure, and neurological, ocular, and skin anomalies. Among the latter, port wine facial birthmarks stand out [62,63,64,65,66]. The etiology of this condition is attributed to somatic mutations that activate genes that encode heterodimeric G-protein chains, specifically the GNB2 gene, tha encodes the beta chain or the GNAQ and GNA11 genes that encode the alpha chain. GNAQ and GNA11 mutations affect mitogen-activated protein kinase (MAPK)/RAS pathway signaling (RAS/MAPK), which is involved in endothelial cell migration. This leads to the formation of leptomeningeal angioma and facial capillary malformation with ipsilateral glaucoma [5,66,67].

These vascular malformations can exhibit mild bone hypertrophy from the ipsilateral cranial vault to the port wine stain, as a compensatory mechanism that addresses the atrophy of the underlying cerebral parenchyma or focal venous hypertension associated with primary venous dysplasia, eventually causing facial distortion and compression effects [67]. Brain ailments can range from a small unilateral parietal–occipital focal area of the brain to extensive bilateral ailments. This results in a variety of symptoms, including seizures, mild learning problems, fine motor deficiencies, vision loss, hemiparesis, strokes that are mainly ischemic, and severe intellectual disability [62,68].

Given the wide variety of manifestations, clinical diagnosis can be difficult. However, electroencephalography studies can help in identifying SWS in newborns by observing background anomalies in the electroencephalogram, which are characterized by low voltage amplitudes and attenuation of dominant rhythms, as well as by acute waves or peak and wave discharges that are associated with clinical or subclinical seizures. Neuroimaging studies are more sensitive and specific in performing the SWS diagnosis, with MRI recommended as a first-line study in the diagnostic approach. Such studies are based on the incidence of facial angioma and leptomeningeal angiomatosis as standards for diagnosis [65,69]. The MRI protocol with a contrast medium, such as gadolinium, facilitates the detection of angiomatosis and the degree of involvement of brain structures. as a second line of study, CT is recommended to identify cerebral calcifications. The CT protocol via single photon emission identifies the decrease in blood flow in the brain area affected by leptomeningeal angiomatosis (Figure 11) [69].

Although there is no specific treatment for SWS, it is paramount to establish and treat SWS-derived epileptic syndrome. Regarding drug treatment options, anticonvulsants, such as topiramate, may cause bilateral acute glaucoma. Currently, there is a consensus that low doses of acetylsalicylic acid may be beneficial, given that they decrease blood flow disturbances and hypoxic-ischemic neuronal lesions [69].

In infants, presymptomatic treatment with anticonvulsant medications and/or aspirin is promising in delaying the onset of seizures, while in patients with refractory epileptic syndromes, the use of neuromodulators, such as cannabidiol, has been suggested. If the location of the lesion is unilateral, hemispherectomy may be used [62]. There are other treatment approaches, such as modulation through bioactive compounds, that induce fibronectin expression or carbonic anhydrase enzyme and acetylcholinesterase, which focus on decreasing the severity of glaucoma. This approach includes sirolimus, which inhibits the mTOR pathway that is characteristically hyperactivated in SWS and other vascular alterations [70].

Hemispherectomy and hemispherotomy are the gold standards for the treatment of SWS related to medically intractable epilepsy. However, several authors reported on good results in seizure control with limited resection (vascular malformation plus adjacent calcified parenchyma plus epileptogenic areas identified by electrocorticography). Most of the papers showed that a partial surgical excision provides for seizure control only for a limited period of time [64]. Compared with extracranial malformation, the ligation of the affluent artery affected by the malformation is sufficient to decrease the flow of the lesion and the depletion of the mass of the malformation [71].

The prognosis of SWS alteration depends on the extent of leptomeningeal angiomatosis and its effect on cerebral perfusion, the severity of ocular involvement, the age of seizure onset, and whether epileptic syndrome is refractory. Damage to neurological functions increases with age and in children with the angiodysplastic variant of SWS, who need regular followup to identify whether malignant transformation of dysplastic tissue exists [67,69,72].

## 4. Discussion

Vascular malformations are structural and functional anomalies of circulatory system vessels. Epidemiology indicates that up to 60% of these malformations are located in the head and neck, generally compromising the nervous system [2,11]. Most of these malformations are linked to alterations in gene expression in connection with the RAS/MAPK/ERK and PI3K/AKT/mTOR pathways. Both are involved in the proliferation, migration, and apoptosis of endothelial cells and other cell lines that are also involved in carcinogenesis processes [5,10,73]. The progress made in the identification and characterization of the various genetic alterations that make up the etiology of vascular malformations has allowed better understanding of their pathophysiology while providing a better outlook for developing approach options that are mainly based on molecules that regulate gene expression and therapies [73].

Relevant examples of such progress include the atorvastatin proposal as a ROCK inhibitor (which leadsg to a decreased incidence of lesions and hemorrhages in CAs [74]), propranolol (which is suggested as an EndMT inhibitor by modulating the TGF pathway and transforming growth factor -β in CA [41,75,76]), or pioglitazone, silibinin, norcantharidine, berberine, protocatechuic aldehyde, hemodine, tuberin, simvastatin, and sirolimus (as regulators of fibronectin expression in SWS [70,73]). In particular, sirolimus has been proposed as a treatment alternative for complicated vascular malformations [9]. Although these drugs have not yet been approved as specific treatments for vascular malformations, they are listed as possible aids against the formation of vascular malformations in multiple familiar cases, or to decrease the likelihood of recurrence after surgical resection or sclerotherapy [76].

The increasing complexity of therapeutic approaches to vascular malformations highlights the need to avoid confusing them with each other through correct characterization. Although clinical data may be useful in this process, such data are not entirely decisive, given that at a clinical level there may be silent courses, as in the cases of CCT [58] and small CAs [31] or clinical manifestations that are closely similar, such headaches and seizures that are present in most of the different vascular malformation cases [17,20,32,33,49,50,51,60,62,68]. An example of a fairly useful clinical feature is port wine stain in the case of SWS [63,64,65]; however, this syndrome will only occur in 8% to 33% of the subjects with the skin mark [70]. In addition, SWS may be present without the presence of port wine stain. Therefore, considering these cases as type III encephalofacial angiomatosis, according to the Roach scale [77], it is not feasible to establish an exclusively clinical diagnosis and it is even less feasible to do so in cases where clinical manifestations are non-specific.

In order to achieve the objective of comprehensively and optimally characterizing vascular malformations, imaging studies are a particularly useful tool. Initially, some entities, such as VMs and AVMs, can be identified through Doppler-type ultrasound [8,15], which allows for the evaluation of morphological and functional data, specifically the low flow that characterizes VMs and the high flow that characterizes AVMs [6,8]. However, this imaging modality is not useful in evaluating deep, large lesions with a bone component [8] or in vascular malformations such as CCT, CA, and leptomeningeal angioma, which associated with SWS. In these cases, the ultrasound does not provide data as conclusive as the data obtained with MRI, which is generally considered to be useful as a first-line imaging study with T2 and T1 protocols generally recommended, except in CA cases in which GRE T2 is recommended because the lesions are angiographically hidden [8,78].

As part of the diagnostic approach, other imaging studies that allow the observation of specific characteristics that go beyond the determination of the presence of vascular malformations need to be considered. They help to characterize the malformations and thus establish guidelines for the most appropriate therapeutic approach. Among the general indicators are the size of the *nidus*, the size of the lesion, the presence or absence of vessels of afferent or efferent circulation, and the location thereof. These indicators emphasize surgical accessibility, and whether it may compromise certain brain areas (e.g., location in an eloquent site) [8]. Other more specific studies allow the establishment of differential diagnoses, including those that discern between CAs and AVMs or CAs and CCT, which can be perceived as being similar according to MRI data. Accordingly, the performance of DSA has been recommended [17,40] to distinguish between CAsand. AVMs, and the performance of SWI has been recommended to distinguish between CAs and CCT [58]. In the case of SWS, the International League Against Epilepsy (ILAE) recommends that a differential diagnosis be established against other epileptic syndromes by taking into account, as highly relevant, the port wine stain data and the leptomeningeal enhancement data obtained via MRI [13,79].

Despite the emerging options for drug treatment for vascular malformations, sclerotherapy, embolization, and surgical resection or laser cytoreduction interventions continue to be the most optimal approach options in most cases, as they allow for the handling of large, surgically accessible lesions; in difficult cases, endovascular sclerotherapy and embolization allow for the treatment of such difficult-to-reach lesions [9,19,70,80]. The exception is CCT or asymptomatic AVMs in older adults, which can generally be managed with a conservative surveillance approach [58,59,80].

As far as endovascular approaches to sclerotherapy and embolization, the high association with lesion recurrence, generally through rechanneling, should be considered. In certain cases, several sessions are required. in addition, these intervention techniques do not allow for the approach of angiographically hidden lesions, as in the case of Cas. For these reasons, the surgical approach can be used in some cases. However, endovascular processes are preferred over surgeries in deep lesions that are difficult to access [17,40,80]. A crucial element to consider in choosing between sclerotherapy and embolization is the risk of dissemination of the sclerosing agent, which can lead to necrosis and embolization of areas that are not directly affected by vascular malformation. This risk increases potentially with high-flow vascular malformation interventions, such as AVMs. A prophylactic measure to avoid this complication is drainage vessel compression of the lesion; however, the use of this measure depends on the location of the malformation and its relationship with other brain structures. Again, this highlights the importance of correctly characterizing the morphology and structural relationship of the vascular malformation via imaging studies [8,81]. Transvenous retrograde nidus sclerotherapy under controlled hypotension (TRENSH) is an embolization variant that consists of the temporary induction of systemic hypotension with or without partial occlusion of feeding vessels. It stands out because of its more complete permeabilization of the *nidus* of malformations with arterial components, which significantly reduces the risks of hemorrhage and embolization in peripheral areas that are unrelated to the vascular malformation [82].

Another approach option in the treatment of vascular malformations, mainly AVMs, is robotic stereotactic radiosurgery (RSR) combined with previous sclerotherapy. RSR has shown encouraging results. Nevertheless, it is a technique that is less performed than surgical resection [52]. Surgical resection is considered to be the intervention technique with the lowest rate of lesion recurrence; however, the high risks of transurgical and post-surgical bleeding during the first year after the intervention must be considered. The Spetzler–Martin scale has been an excellent reference point in guiding whether to choose this approach technique, depending on lesion size, eloquent location, and deep drainage. This underscores how important it is to know the contributions that are made by the different imaging studies in reaching a more accurate estimate of the dimensions of vascular malformations and their anatomical and functional characteristics. This scale also allows us to consider the possibility of establishing multimodal management that consists of endovascular embolization prior to surgery [80]. Although there is a debate between the choice of a surgical approach or sclerotherapy and embolization as the first lines of treatment, there is agreement on the need to adequately characterize the lesions in order to determine the most appropriate management, guided by clinical data, imaging, and histopathological data. The data are mainly gathered during the transoperative period that is associated with the delimitation of resection sites [44,71,78,80,83,84].

Despite the importance of the characterization of vascular malformations in this search, it was difficult to obtain illustrative images of the anatomical and imaging characteristics that had the necessary quality to identify the key differences between each vascular malformation. However, we complemented this review with some original images that allowed us to concentrate and compare the imaging and the anatomical, histopathological, and clinical characteristics.

## 5. Conclusions

The correct anatomical, morphological, and functional characterization of cerebral vascular malformations, via the various imaging studies, is highly relevant in determining a therapeutic approach, either multimodal or isolated. Despite drug advances, theranostic therapy does not exclude the importance of properly characterizing vascular malformations. On the contrary, correct characterizations are useful in evaluating the response to these emerging lines of treatment.

## Figures and Tables

**Figure 1 life-12-01199-f001:**
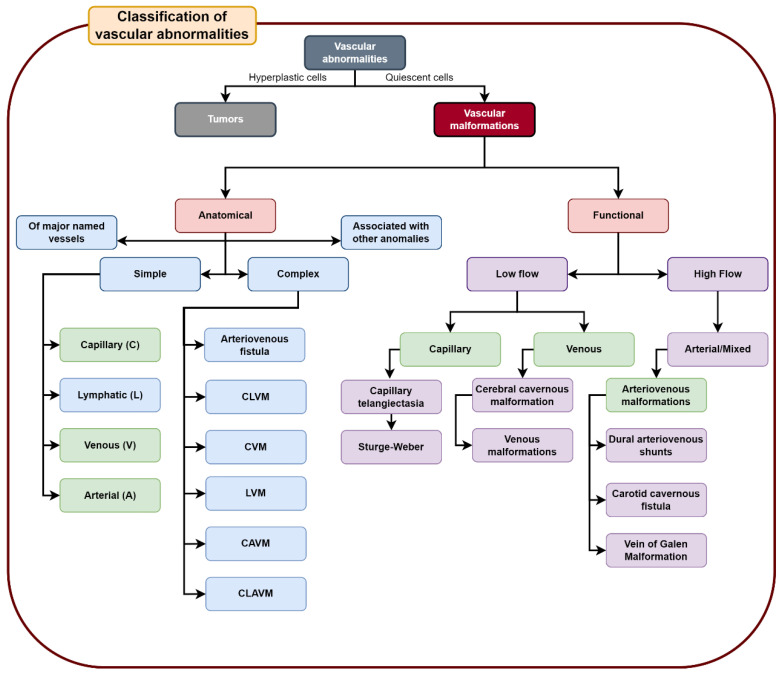
Division of vascular anomalies with a focus on the sub-classifications of vascular malformations. Arteriovenous malformations are categorized in the group of simple malformations, according to ISSVA’s 2018 criteria. Note that the main topics covered in this systematic review are shown in green.

**Figure 2 life-12-01199-f002:**
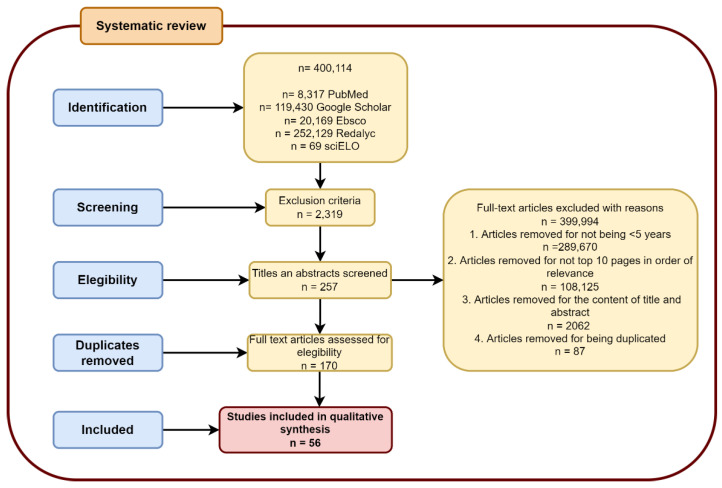
Studies included in the results of this study.

**Figure 3 life-12-01199-f003:**
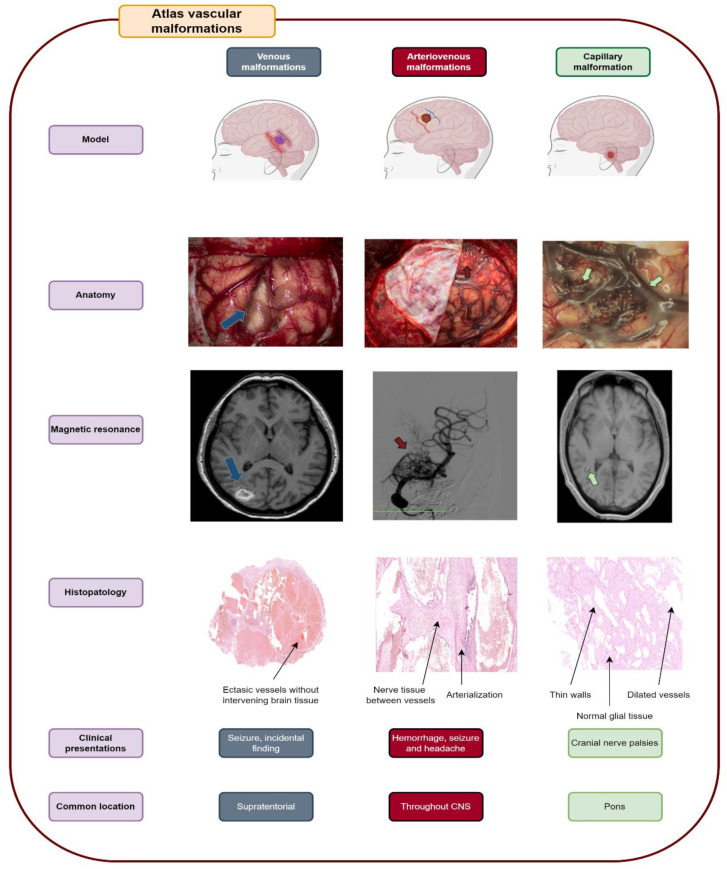
The main topics addressed in this systematic review from the points of view of the model, anatomy, magnetic resonance imaging, histopathology, clinical presentation, and common location [13]. The representative image of venous malformation shown in the figure is cavernoma; the representative image of arteriovenous malformation is a temporal lobe malformation; the representative capillary malformation is capillary telangiectasia.

**Figure 4 life-12-01199-f004:**
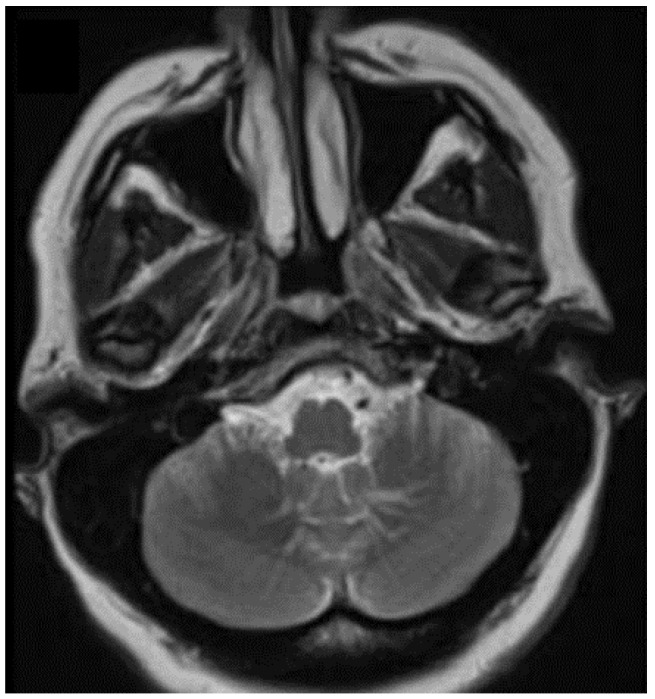
Venous malformation in cerebellar vein, observed by T2 MRI [17].

**Figure 5 life-12-01199-f005:**
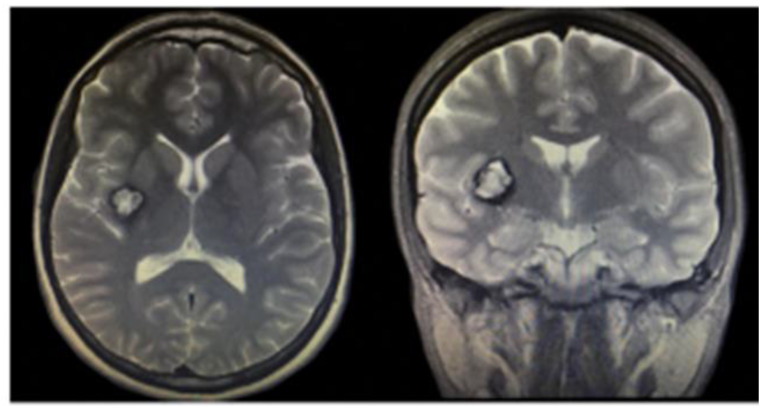
Cavernous angioma in the right insula and putamen. Axial and coronal T2 magnetic resonance imaging [37].

**Figure 6 life-12-01199-f006:**
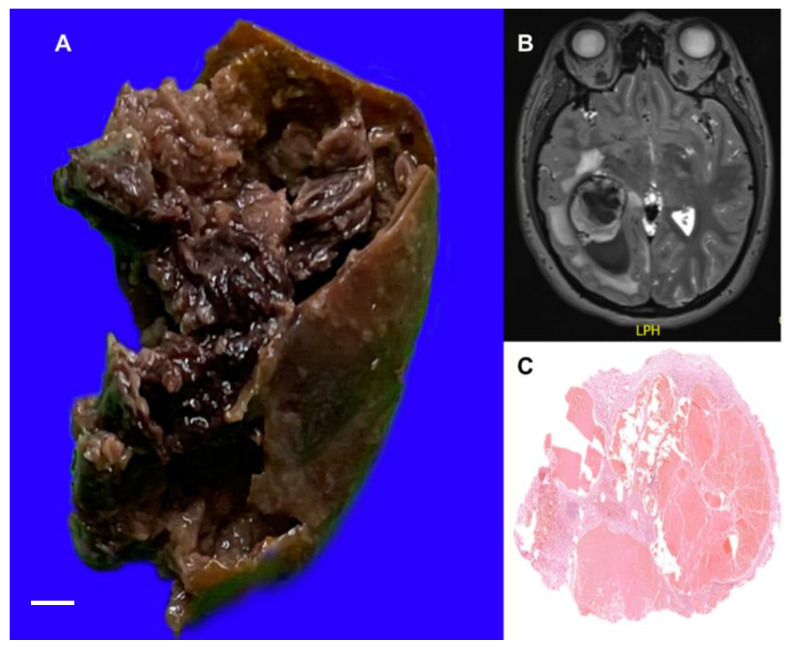
(**A**) Cut surface of cavernous angioma showing cavity with blood debris and fibrous wall. (**B**) Axial T1-weighted magnetic resonance imaging with contrast showing hyperintense oval right parietal lesion in relation to the parenchyma and hypointense center with defined borders and irregular hypointense halo. (**C**) Histologic section of cavernous angioma showing thin-walled ectatic vessels. The white bar represents 1 cm.

**Figure 7 life-12-01199-f007:**
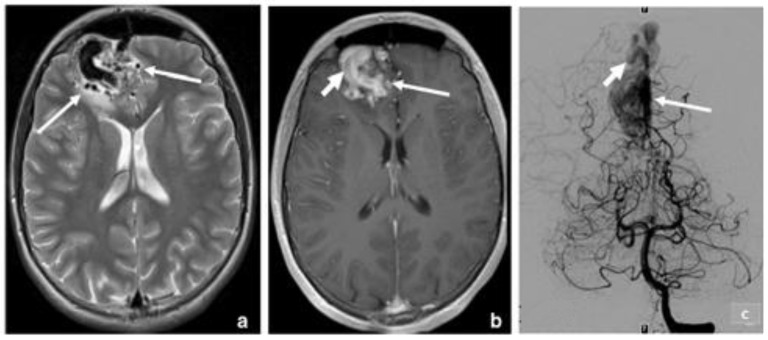
Arteriovenous malformation of the right frontal lobe: (**a**) T2 MRI, axial image, shows multiple hypointense flow gaps, with a slight signal anomaly in the surrounding cerebral parenchyma (arrows); (**b**) with contrast, it shows an early enhancement of AVMs (arrows); (**c**) digital subtraction angiography of the left vertebral artery’s catheterization in the coronal plane shows the AVMs nidus (long arrow) and the drainage veins with opacification (short arrow) [8].

**Figure 8 life-12-01199-f008:**
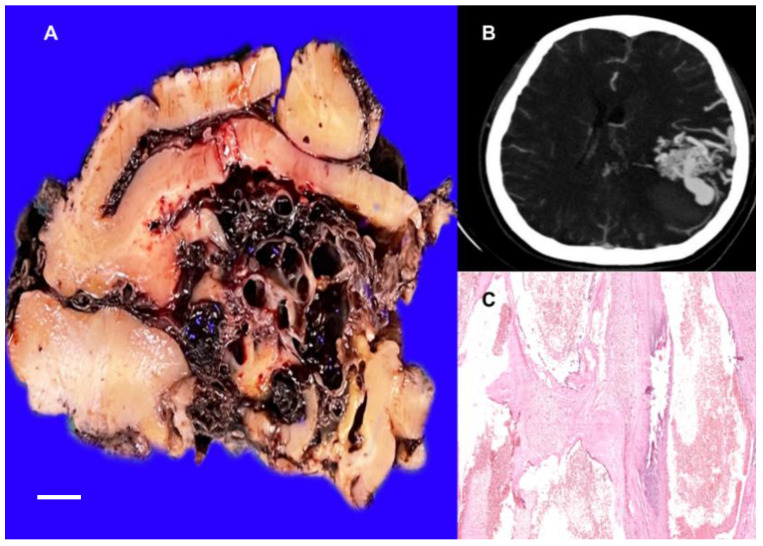
(**A**) Cut surface of arteriovenous malformation showing multiple vessels with variable lumens immersed in brain parenchyma. (**B**) Axial slice CT angiography showing a lesion composed of a vascular conglomerate that enhances the contrast medium. (**C**) Histological section showing vessels with variable lumens and mural thicknesses with arterialization and brain parenchyma between the vessels HE 10×. The white bar represents 1 cm.

**Figure 9 life-12-01199-f009:**
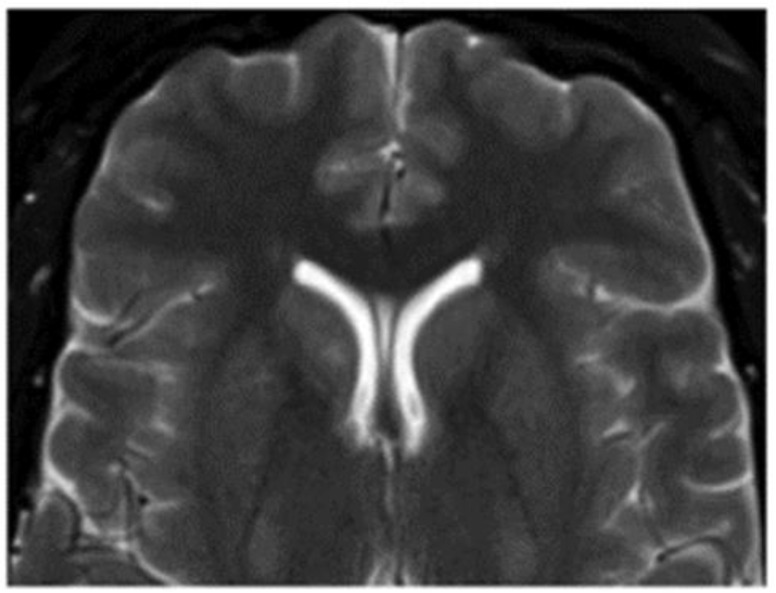
CCT measuring 6 mm in right caudate nucleus, observed through T2 MRI [59].

**Figure 10 life-12-01199-f010:**
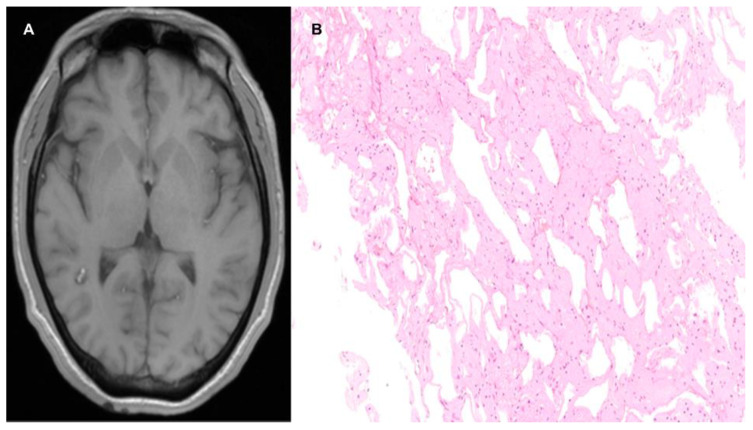
(**A**) Simple T1-weighted axial slice MRI showing an oval lesion in the white matter of the right superior temporal gyrus, with a hyperintense center and a hypointense halo. (**B**) Histologic section shows telangiectasia with small, thin-walled vessels, luminal dilation, and white matter between them.

**Figure 11 life-12-01199-f011:**
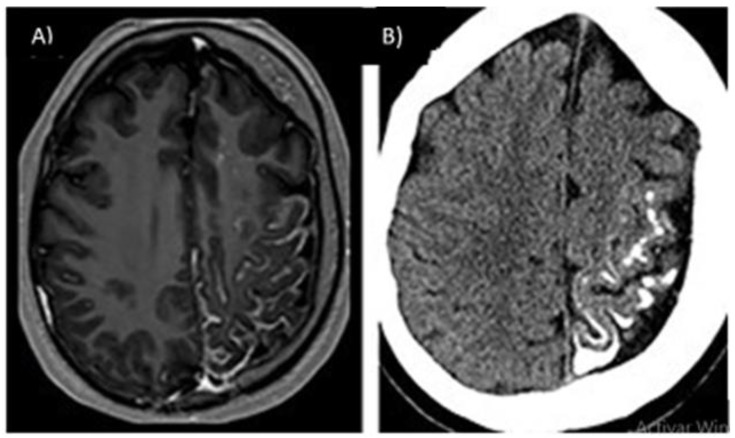
Leptomeningeal angiomas from ipsilateral SWS to port wine lesion (**A**), observed by contrast-enhanced MRI, is shown on the left side in leptomeningeal vascular malformation and gyrate atrophy. (**B**) CT without contrast shows calcifications and twist atrophy [66].

**Table 1 life-12-01199-t001:** Pathophysiology of cavernous angioma according to its location.

Location	Effects
Mesencephalon-pontine	Ataxia, imbalance, multiple falls, headache, blurred vision, and dysarthria [35].
Insular cortex	Sensory, taste and speech symptoms [36].
Insular putamen	Hemichorea–hemiballism [37].
Frontal lobe	Cognitive impairment [38].
Pontocerebellar	Hemifacial spasm, headache [39].

## Data Availability

The review protocol can be accessed to drmarin.neuroscience@gmail.com.

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
