# Peer review of "Atlas of Nervous System Vascular Malformations: A Systematic Review"

_life, 2022, doi:10.3390/life12081199_

Round 1

Reviewer 1 Report

The authors provide an excellent review of the types of vascular malformation of the head and brain. The manuscript is well written and organized and appears to be in a near final state in regards to editing, typesetting, and formatting. There are no glaring omissions and the piece does appear to be comprehensive as claimed. The only suggestion would be to delineate any differences in management between these lesions which may be extracranial and those intracranial. The authors discuss this briefly but this can be better elucidated for each of the malformations in order to help the clinical understand what is possible for diagnosis of each lesion. Otherwise, I feel that the report is nearly complete and would be an excellent inclusion in Life. 

Author Response

Given that most of the cases presented are intracranial, except Sturge-Weber, which is intracranial but also contains an extracranial region, only a section on the extracranial approach of Sturge-Weber was added.

Reviewer 2 Report

Thanks for the opportunity for reviewing this interesting manuscript. This paper is well written and organized. It summarizes brain VMs well. I have no specific comments.

Author Response

Thank you very much. Greetings.